# Structural basis for subtype-specific inhibition of the P2X7 receptor

Akira Karasawa, Toshimitsu Kawate*

Department of Molecular Medicine, Cornell University, Ithaca, United States

**Abstract** The P2X7 receptor is a non-selective cation channel activated by extracellular adenosine triphosphate (ATP). Chronic activation of P2X7 underlies many health problems such as pathologic pain, yet we lack effective antagonists due to poorly understood mechanisms of inhibition. Here we present crystal structures of a mammalian P2X7 receptor complexed with five structurally-unrelated antagonists. Unexpectedly, these drugs all bind to an allosteric site distinct from the ATP-binding pocket in a groove formed between two neighboring subunits. This novel drug-binding pocket accommodates a diversity of small molecules mainly through hydrophobic interactions. Functional assays propose that these compounds allosterically prevent narrowing of the drug-binding pocket and the turret-like architecture during channel opening, which is consistent with a site of action distal to the ATP-binding pocket. These novel mechanistic insights will facilitate the development of P2X7-specific drugs for treating human diseases.

## Introduction

Chronic pain is a major public health epidemic, debilitating more than 10% of adults globally with afflictions like persisting headaches, lower back pain, and rheumatoid arthritis (**Gold and Gebhart, 2010**; **Goldberg and McGee, 2011**). While some patients are responsive to commonly used analgesics, many are resistant to treatment (**Dworkin et al., 2007**; **Finnerup et al., 2010**). The P2X7 receptor, an extracellular ATP-gated ion channel predominantly expressed in immune cells of the blood and the brain (**Surprenant et al., 1996**; **Habermacher et al., 2016**), is an emerging target for treating refractory pain (**Chessell et al., 2005**; **Gum et al., 2012**; **Burnstock, 2013**; **North and Jarvis, 2013**; **Sperlágh and Illes, 2014**). Over the last decade, enormous effort has been made to develop a number of structurally distinct P2X7 specific antagonists, some of which have been demonstrated to alleviate chronic pain in animal models (**Honore et al., 2006a**; **McGaraughty et al., 2007**; **Bartlett et al., 2014**). However, mechanisms of action for these drugs remain poorly understood, hampering the development of effective therapeutic compounds for human patients (**Keystone et al., 2012**; **Stock et al., 2012**; **Bhattacharya and Biber, 2016**).

P2X receptors are trimeric ligand-gated ion channels that facilitate extracellular-ATP mediated signaling along with the G protein-coupled P2Y receptors (**North, 2002**; **Burnstock, 2014**). The P2X7 receptor belongs to the P2X receptor family, however, it was originally identified as a unique ATP-receptor named "the P2Z receptor", as it harbors many characteristics distinct from P2X and P2Y receptors (**Di Virgilio, 1995**). For instance, the P2X7 receptor requires an unusually high concentration of ATP (EC50 $\geq$ 1 mM under physiological ion concentrations) for initial activation (**Yan et al., 2010**), its channel activity is facilitated by prolonged or repeated ATP applications (**Surprenant et al., 1996**), and it opens a membrane pore large enough for molecules up to ~900 Da to permeate (**Steinberg et al., 1987**; **Nuttle and Dubyak, 1994**; **Yan et al., 2008**). While crystal structures of the P2X3 and P2X4 receptors have uncovered common mechanisms such as ATP-binding and gating for the P2X receptor family (**Kawate et al., 2009**; **Hattori and Gouaux, 2012**; **Mansoor et al., 2016**), many questions continue to exist regarding subtype specific mechanisms,

**\*For correspondence:** tk499@ cornell.edu

**Competing interests:** The authors declare that no competing interests exist.

especially for the enigmatic P2X7 receptor subtype. How do structurally-unrelated drugs antagonize only the P2X7 receptor but not the other P2X subtypes? Do they target an activation mechanism unique to the P2X7 subtype? Here we identified and mapped the binding site for the P2X7 specific inhibitors for the first time using X-ray crystallography, and demonstrated by electrophysiological experiments that those inhibitors allosterically abrogate conformational changes associated with P2X7 receptor activation.

## Results

### Architecture of a mammalian P2X7 receptor

To define the structural basis for drug binding, we first sought to determine the crystal structures of a mammalian P2X7 receptor in the presence of five structurally distinct antagonists (A740003 (*Honore et al., 2006a*), A804598 (*Donnelly-Roberts et al., 2009*), AZ10606120 (*Michel et al., 2007*), GW791343 (*Michel et al., 2008a*, *2008b*), and JNJ47965567 (*Bhattacharya et al., 2013*)). Using fluorescence detection size exclusion chromatography (FSEC) (*Kawate and Gouaux, 2006*), we screened full-length and a series of C-terminally truncated versions of nine human P2X7 (hP2X7) orthologues (72–89% identical), and found that an artificially truncated version of the panda (*Ailuropoda melanoleuca*) P2X7 receptor (pdP2X7) not only expressed better than other orthologues in insect cells but also remained trimeric and monodisperse in detergents commonly used for crystallography (*Figure 1A*). Whole cell patch clamp recordings confirmed that pdP2X7 presents comparable characteristics to hP2X7 (*Figure 1B–G*). The truncated pdP2X7 was further optimized to obtain a construct termed pdP2X7cryst($\Delta$1-21/$\Delta$360-600/N241S/N284S/V35A/R125A/E174K) that we used to solve the crystal structures at ~3.5 Å resolution. pdP2X7$_{cryst}$ exhibited slower deactivation and no obvious current facilitation (run-up) after repeated ATP applications (*Figure 1C and F*).

Overall, a single subunit of the P2X7 receptor resembles the 'dolphin-like' shape of zebrafish P2X4 (zfP2X4; 45% identical to pdP2X7) (*Kawate et al., 2009*; *Hattori and Gouaux, 2012*) and human P2X3 (hP2X3; 38% identical to pdP2X7) (*Mansoor et al., 2016*), the other P2X receptor subtypes with known architecture (*Figure 2A* and *Figure 2—figure supplement 1A*). The P2X7 structure obtained in the absence of antagonists (apo-form) likely represents a closed conformation, as the transmembrane helices constrict the channel gate at residues G338, S339, and S342 (*Figure 2B* and *Figure 2—figure supplement 1B*). The structural resemblance of the transmembrane helices between our current P2X7 and the zfP2X4 in the closed state (root-mean-square displacement (RMSD) of the protomers is 2.8 Å) also supports that our apo-structure represents a closed conformation (*Figure 2—figure supplement 1C*). Likewise, antagonist-bound P2X7 structures represent the same closed conformation with a RMSD of less than 0.5 Å to the apo structure, indicating that these drugs likely stabilize a resting closed state.

### A novel drug binding pocket

Surprisingly, all five structurally-unrelated compounds bind in the same pocket formed between neighboring subunits, which is juxtaposed with the ATP-binding pocket (*Figure 2C–D* and *Figure 2—figure supplement 2*). This drug-binding pocket is surrounded by thirteen residues projecting mainly from $\beta$-strands ($\beta$4, $\beta$13 and $\beta$14) in the upper body domains of the neighboring subunits (*Figure 3A*). While the precise distances and angles between the side-chains and the drugs cannot be determined at the current resolutions (~3.2–3.6 Å), electron density was clear enough to localize and orient the side chains of the drug-coordinating residues (*Figure 3—figure supplement 1* and *Video 1*). Drug binding seems to be mediated mainly by hydrophobic interactions, especially at positions deep within the cavity, involving F95, F103, M105, F293, and V312. Despite structural diversity, all five P2X7 antagonists fit within the drug-binding pocket (*Figure 3B*), highlighting that the size and the shape of the drug-binding pocket play major roles in determining the affinity and specificity of the drugs. Indeed, the equivalent pocket in the P2X4 receptor is too narrow to accommodate the smallest P2X7 antagonist, A804598, even though it is similarly hydrophobic to that of the P2X7 receptor (*Figure 3—figure supplements 2* and *3*). Thus, we suggest that the differences in the size of the inter-subunit hydrophobic pocket is the major factor that confers P2X7 specific binding of the inhibitors.

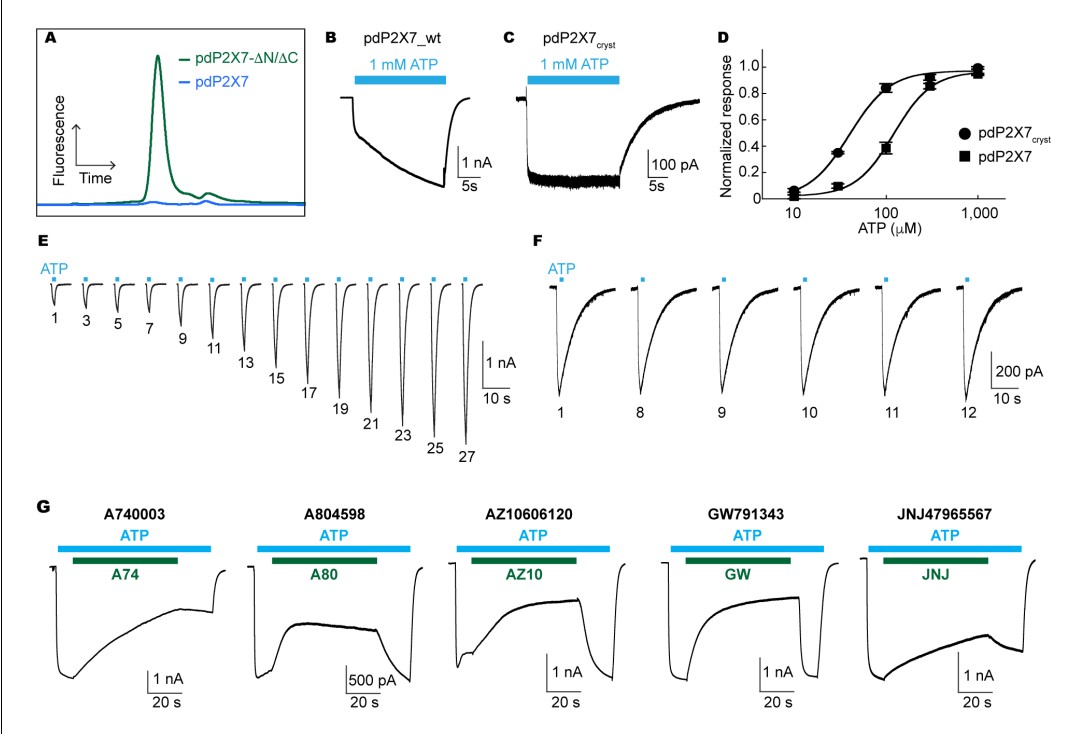

**Figure 1.** Characterization of pdP2X7. (**A**) FSEC traces (Ex: 488 nm and Em: 509 nm) for the full length (blue) and the truncated (green) pdP2X7. (**B**) and (**C**) Whole cell patch clamp recordings from the wildtype pdP2X7 (**B**) and pdP2X7$_{cryst}$ (**C**) triggered by 1 mM ATP. (**D**) ATP dose responses of pdP2X7 (square) and pdP2X7$_{cryst}$ (circle) determined by the whole cell patch clamp experiments. The plots were made using the normalized means of five independent experiments and the error bars represent SEM. Dose response curves were fit with the Hill equation. EC50 values of pdP2X7 and pdP2X7$_{cryst}$ were 122 μM and 40 μM, respectively. (**E**) and (**F**) Whole cell patch clamp recordings from pdP2X7 (**E**) and pdP2X7$_{cryst}$ (**F**) expressed in HEK293 cells. Each trace represents the pdP2X7 mediated current triggered by 1 s applications of 100 μM ATP from the same patch. The numbers below the traces indicate the number of repeated ATP applications. (**G**) Whole cell patch clamp recordings of pdP2X7 triggered by 1 mM ATP in the presence of different P2X7 specific antagonists. Drugs were applied for 1 min in the presence of ATP. Membrane was held at −60 mV. Concentrations of the drugs were; A740003: 600 nM; A804598: 180 nM; AZ10606120: 2.3 μM; GW791343: 50 μM; JNJ47965567: 136 nM.

We further validated the drug-receptor interactions by site-directed mutagenesis on the drug-binding residues. To facilitate robust and systematic data collection, we monitored cellular uptake of the fluorescent dye, YO-PRO-1, as a proxy for receptor activity (*Surprenant et al., 1996*) (*Figure 3—figure supplement 4A*). Consistent with the crystal structures, the mutants, F88A, F95A, F103A, M105A, F108A, and V312A, showed increased IC50 values (see *Table 1* for IC50 values on the wildtype), supporting that these residues play important roles in drug binding (*Figure 3C* and *Figure 3—figure supplement 4B*). In particular, interaction with F103 is crucial for the inhibitory action of all five drugs.

## Allosteric non-competitive inhibition

Binding of all five drugs to a site distinct from the ligand-binding pocket suggests that these compounds act as non-competitive inhibitors. However, previous studies using cell-based $Ca^{2+}$ influx and IL-1$\beta$ release assays proposed that three of those compounds, A740003, A804598, and JNJ47965567, compete against ATP-binding to inhibit the P2X7 receptor (*Honore et al., 2006a*; *Donnelly-Roberts et al., 2009*; *Bhattacharya et al., 2013*). To clarify the working mechanism for each drug, we measured the dose responses of the P2X7 mediated YO-PRO-1 uptake in the presence of each drug at multiple concentrations. We used BzATP—an artificial but potent agonist of P2X7 receptors—for these experiments to achieve saturating responses in the presence of concentrated antagonist. For each drug, dose response curves fit well with a non-competitive inhibition model (*Kenakin, 2006*), but poorly with a competitive model (*Figure 3—figure supplement 5*).

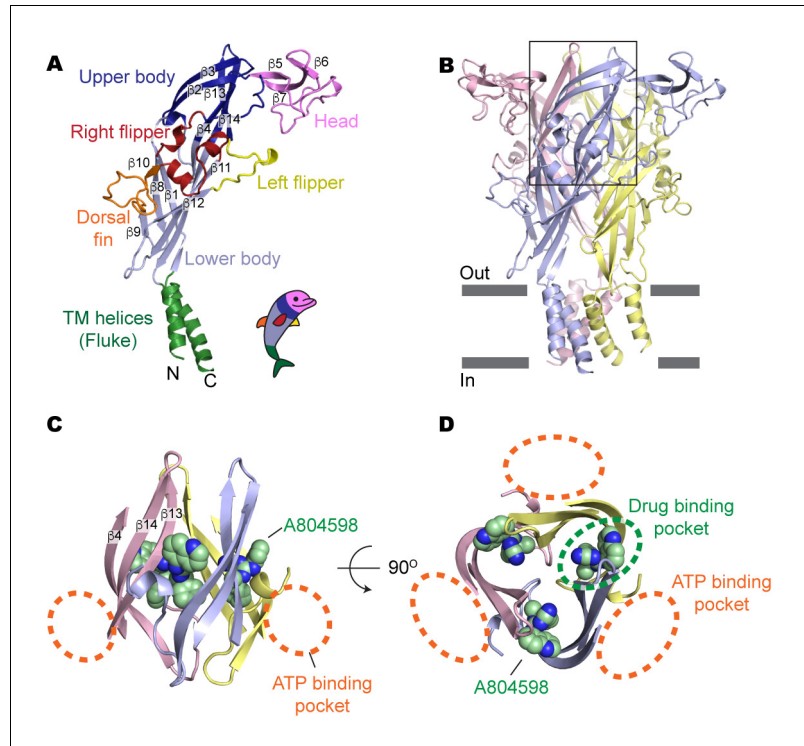

**Figure 2.** Drug-binding pocket of the P2X7 receptor. (**A**) Cartoon representation of a 'dolphin-like' single subunit of the apo pdP2X7 structure. Fourteen beta strands are labeled as β1-14. Each domain is colored consistent with the previous studies for better comparison (*Kawate et al., 2009*; *Hattori and Gouaux, 2012*). (**B**) Cartoon representation of the trimeric pdP2X7 structure viewed from the side. The black box indicates an approximate location of the upper body domains shown in (**C**) and (**D**). (**C**) Side view of the upper body domains exhibiting A804598 binding sites with respect to the ATP-binding pockets (orange dashed lines). A804598 is shown as CPK spheres. (**D**) Top view of the apo pdP2X7 structure with respect to the ATP-binding pockets (orange dashed lines) and one of the drug-binding pockets (green dashed line).

The following figure supplements are available for figure 2:

**Figure supplement 1.** Structural comparison between pdP2X7 and zfP2X4.
**Figure supplement 2.** Electron density around the P2X7 specific antagonists.

Furthermore, the Schild plots displayed non-linear relationships, especially at higher concentrations, consistent with a non-competitive mechanism (*Figure 3D*) (*Schild, 1947*).

To confirm the non-competitive mode of inhibition, we performed an ATP-binding assay on purified P2X7 receptor pretreated with each drug. We exploited fluorescence anisotropy (*Figure 3—figure supplement 6A–C*) using a fluorescently-labeled ATP analogue (Alexa-ATP), which is as potent as ATP and is capable of triggering P2X7 channel opening at 10 µM in the absence of divalent cations (*Figure 3E*). At this concentration, Alexa-ATP gave rise to fluorescence anisotropy when incubated with P2X7 in a dose-dependent manner (*Figure 3—figure supplement 6D*). Notably, drug-treated P2X7 receptor exhibited similar levels of fluorescence anisotropy, which drastically decreased in the presence of unlabeled-ATP (*Figure 3F* and *Figure 3—figure supplement 6E*). Together, these experiments strongly support that all five P2X7 antagonists are allosteric non-competitive inhibitors.

## P2X7 specific conformational change during channel activation

We propose that the five studied drugs antagonize the P2X7 receptor through a common mechanism and that the unique inter-subunit cavity may be a critical locus for functional regulation.

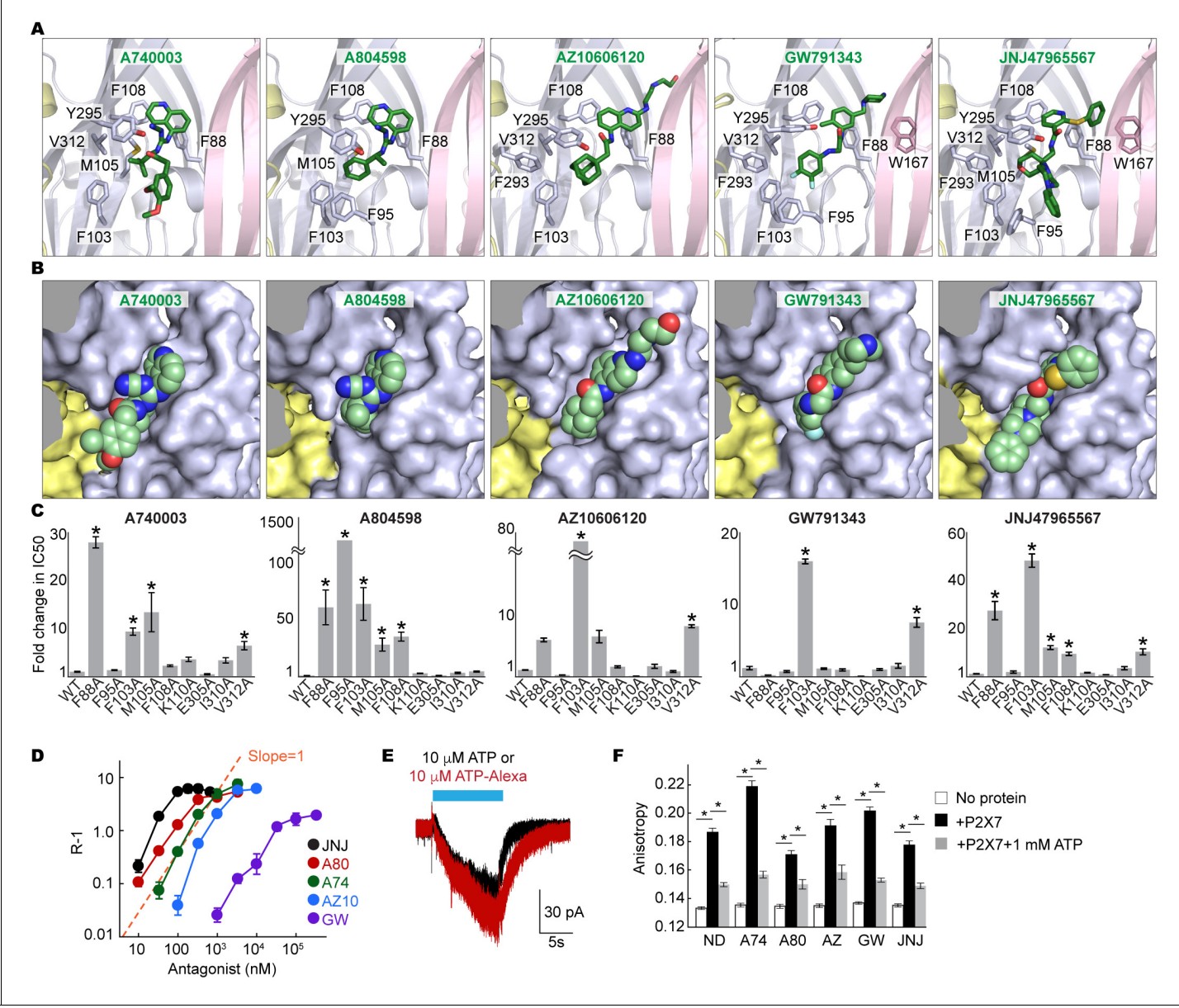

**Figure 3.** Coordination of the P2X7 specific antagonists. (A) Stick representations of the indicated P2X7 specific antagonists (green) and the binding residues in the drug-binding pocket. Structural frame of each subunit is depicted as a cartoon representation in a different color. Oxygen atoms are shown in red and nitrogen atoms are shown in blue. (B) Surface representation of the drug-binding pocket with CPK sphere representation of each drug (green). Yellow and light blue surfaces represent two of the three subunits. The third subunit was omitted for clear representation. (C) Fold changes in IC50 values of the P2X7 specific inhibitors against the YO-PRO-1 uptake activity on alanine mutants of the drug coordinating residues. Increased IC50 values in some of the alanine mutants indicate that these residues bind to the P2X7 antagonists. (D) Schild plots against drug concentrations over a range of three orders of magnitude. Plots for all five of the different P2X7 specific antagonists are non-linear, supporting that these drugs work non-competitively. (E) Whole cell patch clamp recordings of the wildtype pdP2X7 triggered by 10 μM ATP (black) or 10 μM ATP-Alexa (red). The holding potential was −60 mV. (F) Fluorescence anisotropy derived from 10 μM ATP-Alexa without protein (white), with pdP2X7 (black), or with pdP2X7 in the presence of 1 mM ATP (grey). The concentration of pdP2X7 was 100 μM. Concentrations of the drugs were: A740003: 600 nM; A804598: 180 nM; AZ10606120: 2.3 μM; GW791343: 50 μM; JNJ47965567: 136 nM. The dots and the bars represent the means of five independent experiments and the error bars represent SEM. Asterisks indicate significant differences from wildtype or the no protein control (p<0.01) determined by one way ANOVA followed by Dunnett's test.

The following figure supplements are available for figure 3:

**Figure supplement 1.** Side-chain electron density near the drug-binding pocket.

*Figure 3 continued*

**Figure supplement 2.** The equivalent position of the drug-binding pocket in the P2X4 receptor is too narrow for P2X7 antagonist to fit in.

**Figure supplement 3.** Sequence alignment of P2X7 and P2X4 receptors.

**Figure supplement 4.** YO-PRO-1 uptake assays on the alanine mutants of the drug coordinating residues.

**Figure supplement 5.** The P2X7 specific antagonists are allosteric non-competitive inhibitors.

**Figure supplement 6.** Fluorescent anisotropy experiments.

Interestingly, the inter-subunit cavity formed by β13 and β14 in the upper body domain is much wider in P2X7 than in zfP2X4 or in hP2X3 (*Figure 4A–C*) (*Mansoor et al., 2016*). Furthermore, this 'turret-like' structure and the cleft corresponding to the P2X7 drug-binding pocket remain relatively occluded in zfP2X4 and in hP2X3 after activation by ATP (*Figure 4A–C* and *Figure 4—figure supplement 1*) (*Mansoor et al., 2016*). Do the drug-binding pocket and the turret in P2X7 become narrower during channel activation? To explore the involvement of the inter-subunit cavity in P2X7 receptor activation, we monitored the movement of the cavity residues with and without ATP. We first created a series of single cysteine mutants in the drug-binding pocket (*Figure 4D*) and measured ATP-gated channel activity after applying a large cysteine-reactive agent (MTS-TPAE; Mw: 447 Da). We reasoned that modification of a cysteine residue with a bulky moiety should interfere with the conformational changes necessary for channel opening, thereby resulting in diminished channel activity. When MTS-TPAE was applied in the absence of ATP, four cysteine mutants (F103C, K110C, T308C, and I310C) showed irreversible current reduction (*Figure 4E and F*), consistent with the idea that the covalently bound MTS-TPAE at these positions hinders the conformational changes required for channel opening. When MTS-TPAE was applied in the presence of ATP, on the other hand, none of the cysteine mutants presented significant current reduction (*Figure 4E and F*). These results indicate that at least four residues in the drug-binding pocket are more accessible to MTS-TPAE in the closed state than in the open state. Our data therefore suggest that the drug-binding pocket narrows upon ATP binding and that such a conformational rearrangement is crucial for P2X7 channel opening.

To examine the movement of the turret during P2X7 activation, we took advantage of a cysteine mutation at Y295, whose side chain faces the center of the turret (*Figure 5A*). Because two of the three introduced cysteines may form a disulfide bond, we pretreated the cells with a reducing agent prior to cysteine accessibility experiments. Under these conditions, MTS-TPAE diminished the Y295C channel activity by ~80% in the absence of ATP but to a lesser extent (~40%) in the presence of ATP (*Figure 5B and C*). These data support that the turret also narrows during P2X7 activation. Widening of the turret upon ATP-binding is unlikely, as 1) MTS-TPAE had no effect on the Y295C mutant prior to treatment with a reducing agent (*Figure 5D*), indicating the formation of a disulfide bond that would bring the two neighboring subunits closer (Cβ-Cβ distance would change from ~14 Å to ~4 Å) and 2) Y295C mutant exhibited comparable current density with the wildtype in the absence of a reducing agent (*Figure 5—figure supplement 1A*). Notably, narrowing of the inter-

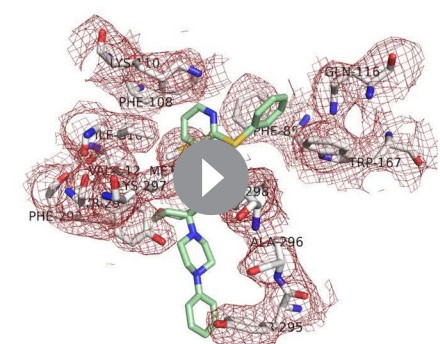

**Video 1.** Electron density of the drug-coordinating residues. 2Fo-Fc electron density map of the JNJ47965567-bound P2X7 structure contoured at σ = 1.0. Electron density is shown as mesh in firebrick red. JNJ47965567 (green) and amino acid residues (white) are depicted as stick representations. Red represents oxygen and blue represents nitrogen.

**Table 1.** IC50 value of each P2X7 inhibitor from the YO-PRO-1 uptake assay.

| Inhibitor | A740003 | A804598 | AZ10606120 | GW791343 | JNJ47965567 |
|---|---|---|---|---|---|
| IC50 | 69.3 nM | 21.7 nM | 231 nM | 8.9 µM | 11.9 nM |

subunit space seems unique to the P2X7 receptor, as suggested by the crystal structures of P2X3 and P2X4 solved in the presence of ATP. Indeed, MTS-TPAE application either in the presence or absence of ATP did not exhibit current reduction for cysteine mutants of the counterpart residues in the P2X4 receptor (*Figure 5B and E*, *Figure 5—figure supplement 1B and C*).

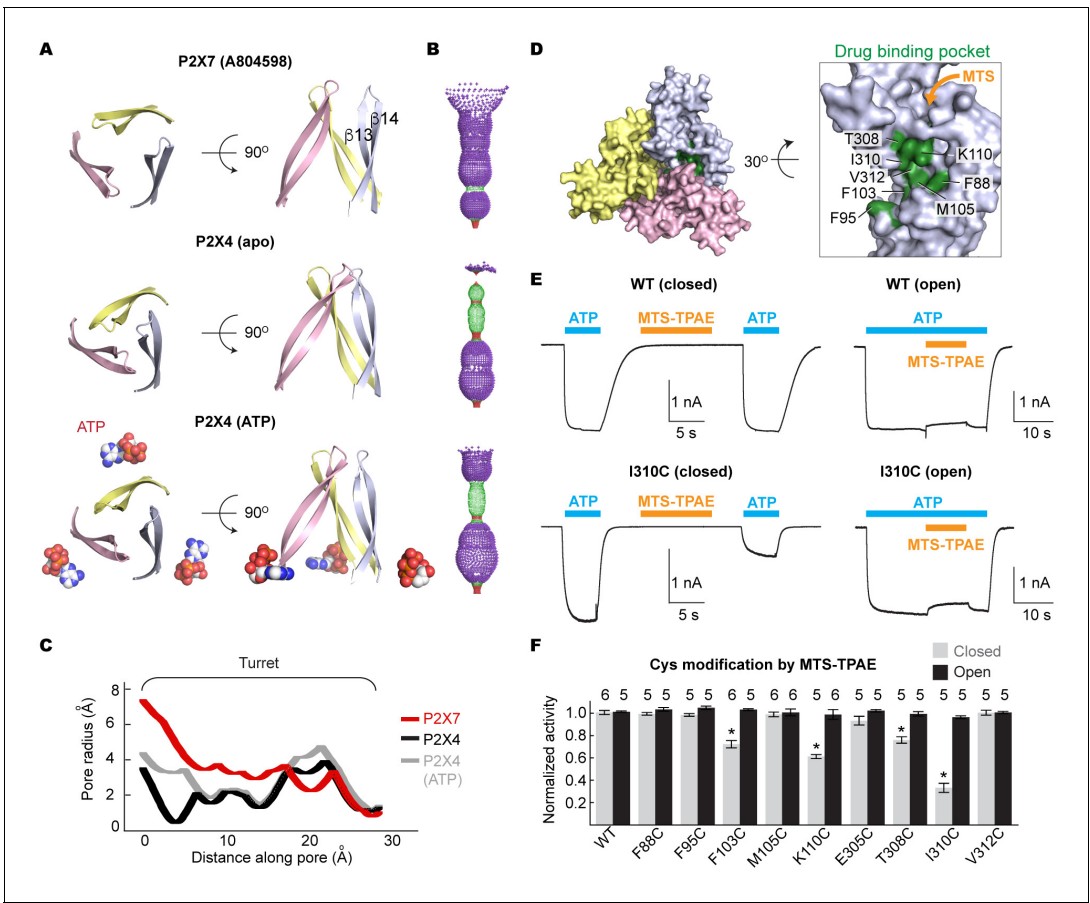

**Figure 4.** Drug-binding pocket narrows during P2X7 activation. (**A**) Cartoon representation of the turret formed by β13 and β14. P2X7 (apo), zfP2X4 (apo), and ATP-bound zfP2X4 are shown as the top (left) and the side (right) views. (**B**) Dot representations of the internal-space along the molecular threefold axis running through the center of the apo closed pdP2X7 (top), the apo closed zfP2X4 (4DW0; middle), and the ATP-bound zfP2X4 (4DW1; bottom). The dot plots are made using the HOLE program (*Smart et al., 1996*). Purple: > 2.3 Å; green: 1.15–2.3 Å; red: < 1.15 Å. (**C**) Central pore radii of the three P2X crystal structures shown in (**B**). (**D**) Surface representation of the residues in the drug-binding pocket tested for accessibility. (**E**) Whole cell patch clamp recordings of the wildtype pdP2X7 and the I310C mutant triggered by 1 mM ATP. MTS-TPAE (1 mM) was applied in the absence (left) or presence (right) of ATP for 10 s to probe the accessibility in the closed or open state, respectively. The membrane was held at −60 mV. (**F**) Normalized channel activities after MTS-TPAE treatment during the closed (grey) or open (black) states. The bars represent the means of five or more independent experiments (numbers above the bars indicate the n value) and the error bars represent SEM. Asterisks indicate significant differences from the widltype control (p<0.01) determined by one way ANOVA followed by Dunnett's test.

The following figure supplement is available for figure 4:

**Figure supplement 1.** Volume of the P2X7 drug-binding pocket is larger than the corresponding clefts in P2X3 or P2X4.

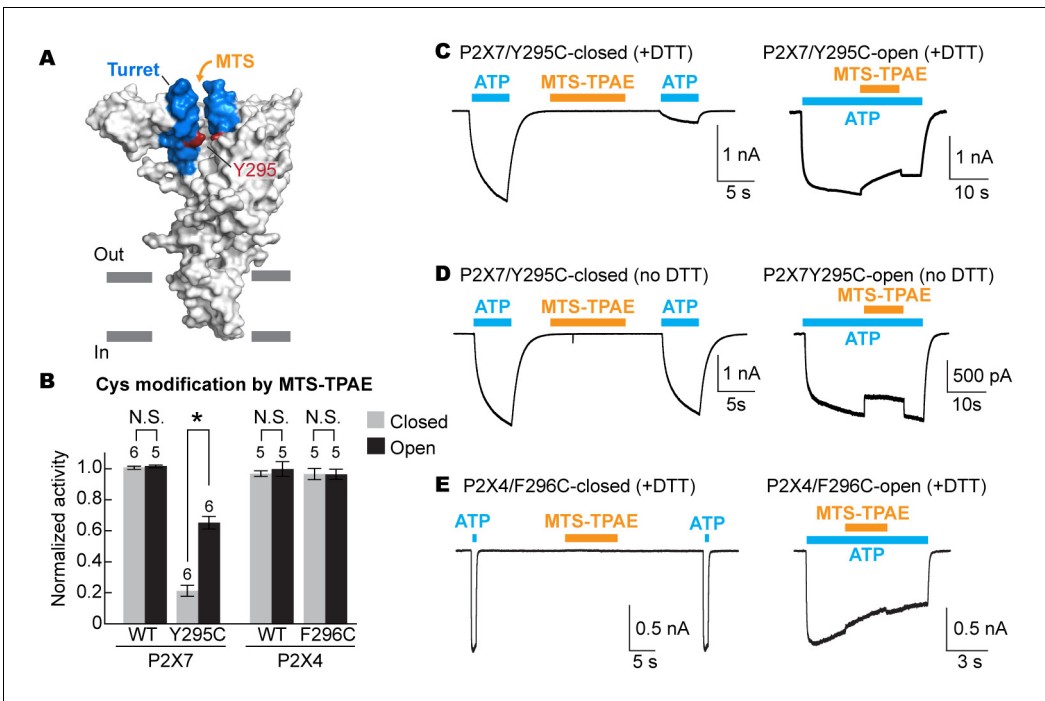

**Figure 5.** The turret closes during P2X7 activation. (**A**) Surface representation of the P2X7 receptor viewed from the side highlighting the location of Y295 (red) facing the center of the turret (blue). Only two of the three subunits are shown for clarity. (**B**) Summary of MTS-TPAE accessibility. Normalized channel activities after MTS-TPAE treatment indicate that Y295C in P2X7 is more accessible in the closed state than in the open state. The bars represent the means of more than five independent experiments (numbers above the bars indicate the n value) and the error bars represent SEM. Asterisk indicates significant difference in normalized channel activity between the cells treated with MTS-TPAE in the closed and in the open states (p<0.01) determined by Dunnett's test. N.S. indicates not significant. (**C**) and (**D**) Whole cell patch clamp recordings of the P2X7/Y295C mutant triggered by 1 mM ATP. MTS-TPAE (100 µM) was applied in the absence (left) or presence (right) of ATP for probing the accessibility in the closed or open state, respectively. Cells were used before (**D**) or after treating with 10 mM DTT for 5 min (**C**). Membrane was held at −60 mV. (**E**) Whole cell patch clamp recordings of the P2X4/F296C mutant triggered by 10 µM ATP. Cells were treated with 10 mM DTT for 5 min before recordings.

The following figure supplement is available for figure 5:

**Figure supplement 1.** The equivalent position of the drug-binding pocket in the P2X4 receptor is not accessible.

---

To confirm that allosterically-bound P2X7 inhibitors prevent the turret from narrowing upon ATP-binding, we obtained the crystal structure of the P2X7 receptor in the presence of both ATP and A804598 (*Figure 6A–D* and *Figure 6—figure supplement 1*). Consistent with the activation mechanisms proposed for P2X3 and P2X4, ATP-binding brings the dorsal fin domain towards the head domain and pushes the left flipper domain away from the ATP-binding pocket (*Figure 6C*). These movements are coupled with widening of the lower body domain (*Figure 6D*), though to an extent in which the transmembrane helices remain closed. In contrast, little conformational change was observed for the upper body domain including the turret and the drug-binding pockets (*Figure 6A*), supporting that the turret closure is essential for P2X7 channel opening. Altogether, our data suggest that P2X7 receptors undergo unique conformational rearrangements where both the drug-binding pocket and the turret in the P2X7 receptor narrow upon ATP-binding. Because such conformational changes are required for channel opening, binding of the P2X7 antagonists preclude these constrictions, thereby efficiently blocking receptor activation (*Figure 7* and *Video 2*).

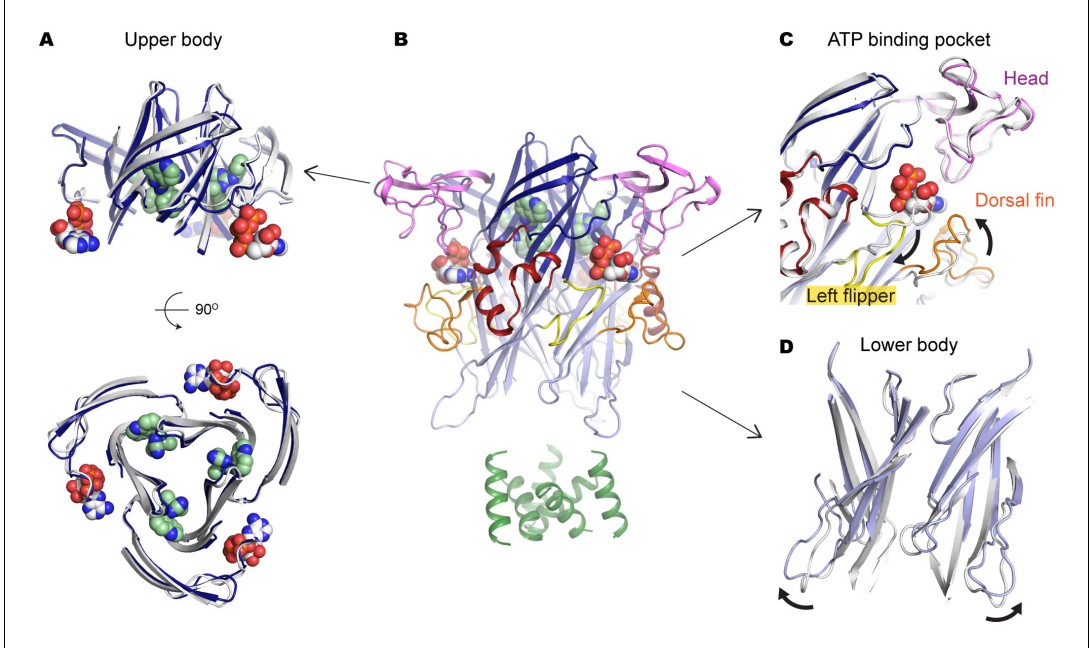

**Figure 6.** A804598 prevents conformational changes in the upper body domain triggered by ATP-binding. (**A**) Cartoon representations indicate little conformational change upon ATP-binding in the upper body domain. Each domain of the ATP/A804598-bound P2X7 is colored according to *Figure 2* and the overall structure is shown in (**B**). Apo P2X7 structure is presented in gray for comparison (**A**, **C** and **D**). (**C**) and (**D**) ATP-binding evokes conformational rearrangement in the ATP-binding pocket (**C**) and in the lower body domain (**D**). The arrows highlight the direction of the domain movement upon ATP binding.

The following figure supplement is available for figure 6:

**Figure supplement 1.** P2X7 structure in the ATP/A804598-bound state.

## Discussion

The presented crystal structures uncover the unique inter-subunit cavity in the upper body domain of the P2X7 receptor. Our data suggest that this cavity shrinks during activation, which allows the lower body domain to widen further. Indeed, the ATP-binding left flipper domain is connected with the pore-lining transmembrane helix through the turret composed of two β-strands (β13 and β14), which supports that movements of the upper body domain are tightly coupled with the channel

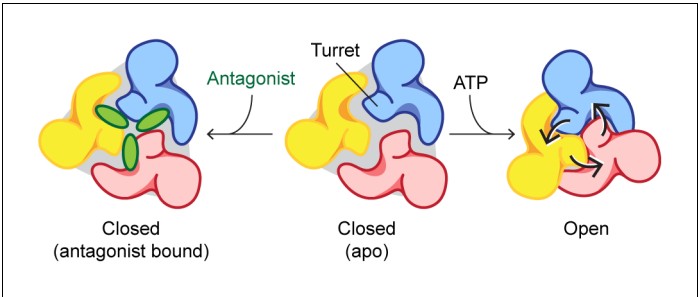

**Figure 7.** Mechanisms of P2X7 activation and inhibition. Schematic representations of the P2X7 receptor viewed from the top. Each color represents a different subunit. The drug-binding pocket and the turret narrow during channel activation. P2X7 specific antagonists stabilize its closed conformation by preventing the movement of these inter-subunit cavities.

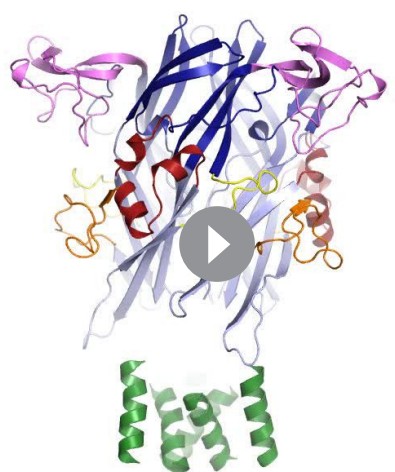

**Video 2.** Conformational differences between the apo and ATP/A804598-bound P2X7 structures. Each domain of the ATP/A804598-bound P2X7 is colored according to *Figure 2*. ATP and A804598 molecules are presented as CPK spheres.

opening (*Figure 2A*). Although the lower body domain in our ATP/A804598-bound structure does seem to widen to some extent upon ATP-binding, the turret diameter remains locked due to A804598 binding, rendering the accompanying movement of the pore-lining transmembrane helices insufficient for channel opening (*Figure 6* and *Video 2*). Closure of the turret, therefore, seems to be a prerequisite for full-widening of the lower body domain, which is necessary for channel opening. Interestingly, the turrets of both P2X3 and P2X4 are narrower even in the closed state, which may facilitate subsequent conformational changes in the lower body domain. Obviously, the crystal structure of an ATP-bound, open-state P2X7 would strengthen our hypothesis. However, it is technically challenging to obtain such a crystal structure, as ATP-bound P2X7 receptors are prone to aggregation in a detergent containing solution. Future efforts to uncover molecular details about the open state of P2X7 may yield further insight into subtype-specific activation mechanism.

Our data demonstrate that all five tested drugs are allosteric and non-competitive inhibitors, which is in contrast to the previous studies showing that A740003, A804598, and JNJ47965567 are competitive inhibitors (*Honore et al., 2006b*; *Donnelly-Roberts et al., 2009*; *Bhattacharya et al., 2013*). This discrepancy may be attributed to the high agonist concentrations (i.e. up to 3 mM ATP or 1 mM BzATP) required for obtaining reliable EC50 values in the presence of antagonists. With such a high concentration of agonists, we noticed that the YO-PRO-1 uptake activity starts to decline ~10 min after agonist application. To minimize potential experimental artifacts, we took advantage of the initial rates instead of total counts at a certain time point used by the previous studies. Nevertheless, the Schild plots of these three compounds appear to be linear at lower concentrations (*Figure 3D*), making it difficult to rely solely on the Schild regression analysis for determining the mode of drug action. We therefore provided extra evidence of non-competitive allosteric drug action using 1) curve fitting with known models of competitive and non-competitive inhibition, 2) ligand-binding assays in the presence of antagonists, and 3) the crystal structure of P2X7 obtained in the presence of both an agonist and an antagonist.

Combined with the crystal structures, information about the drug-receptor specific interactions provide an excellent platform for rationally improving physicochemical properties of currently available P2X7 antagonists. For example, replacing the phenyl group in JNJ47965567 with a slightly bulkier group may increase the binding affinity of this compound, as that would fill the void around residue F95. Likewise, low potency of AZ10606120 and GW791343 (*Table 1*) may be overcome by replacing chemical moieties with those fit better into the drug-binding pocket. Our data also suggest that it may be possible to attach an extra chemical group to the site away from the residue interacting with F103 without affecting binding of the P2X7 antagonists. This leads to an interesting possibility that these drugs could be conjugated with cell penetrating peptides or specific ligands expressed on endothelial cells, which may improve the delivery of drugs across the blood-brain barrier (*Chen and Liu, 2012*). In addition, identification of an unexpected binding pocket within P2X7, that is absent in the closely related P2X4, should allow for engineering of isoform-specific drugs. Development of a new generation of P2X7 antagonists with increased clinical efficacy has the potential to have a significant impact on the treatment of chronic pain and inflammation.

# Materials and methods

## Construct design

Ten mammalian P2X7 genes (human: Q99572.4; giant panda: XP_002913164.1; horse: XP_001495622.1; cattle: NP_001193445.1; dog: NP_001106927.1; rabbit: XP_002719791.1; rat: Q64663.1; mouse: Q9Z1M0.2; opossum: XP_001373213.1; guinea pig: NP_001166578.1) were synthesized based on the protein sequences (GenScript, Piscataway, NJ) and subcloned into a modified pFastBac baculovirus expression vector (Thermo Fisher Scientific, Waltham, MA) that harbors a Strep-tag and EGFP at the N-terminus (pNGFP-FB3). These P2X7 constructs were expressed in Sf9 insect cells and screened using the FSEC strategy (*Kawate and Gouaux, 2006*). Briefly, Sf9 cells at a cell density of $1.0 \times 10^6$ cells/mL in six well plates were infected with 20 µl/well of P1 virus ($10^6$–$10^7$ PFU/mL), harvested after 48 hr, and solubilized in 150 µl of S1 buffer (1% n-Dodecyl-$\beta$-D-Maltopyranoside (DDM; Anatrace, Maumee, OH) and Halt protease inhibitor cocktail (Thermo Fisher Scientific) in phosphate-buffered saline; (pH 7.4)) for 30 min at 4°C. The soluble fractions were collected by ultra-centrifugation at $87,000 \times g$ for 30 min and analyzed by FSEC. Because the full-length P2X7 receptors were poorly expressed in Sf9 cells and did not crystallize in our initial trials, C-terminally truncated versions of the nine P2X7 receptor orthologues (~240 residues removed from the C-terminus) were rescreened by FSEC. Based on the sharp and symmetrical peak profile, the C-terminally truncated panda P2X7 (pdP2X7) was selected for crystallization trials. A series of terminal deletions and glycosylation mutants of pdP2X7 were screened for improving the expression level, stability, and monodispersity. The best behaved construct ($\Delta$1-21/$\Delta$360-600/N241S/N284S) formed crystals in the P4$_2$ space group that diffracted to ~7 Å, which was sufficient for obtaining an initial electron density map by molecular replacement using the crystal structure of the zebrafish P2X4. To improve crystal packing, we screened combinations of point mutants at the predicted crystal contact sites for crystallization/diffraction behavior. Finally, we obtained a construct termed pdP2X7$_{cryst}$ ($\Delta$1-21/$\Delta$360-600/N241S/N284S/V35A/R125A/E174K), which formed crystals in a different space group (I2$_1$3) that diffracted to ~3.5 Å.

## Expression and purification

A detailed protocol for expression and purification of pdP2X7 is available at Bio-protocol (*Karasawa and Kawate, 2017*). pdP2X7$_{cryst}$ was expressed as an EGFP-fusion protein using a baculovirus-insect cell system. Sf9 cells were infected at $3.5$–$4.0 \times 10^6$ cell/mL with 25–30 mL/L P2 virus ($10^7$–$10^8$ PFU/mL) for one day at 27°C and for another two days at 18°C. Cells were harvested by centrifugation at $2040 \times g$, washed with PBS, and resuspended with PBS containing leupeptin (0.5 µg/mL), aprotinin (2 µg/mL), pepstatin A (0.5 µg/mL), and phenylmethylsulfonyl fluoride (0.5 mM). The cells were broken by nitrogen cavitation at 600 psi using a 4635 cell disruption vessel (Parr Instrument, Moline, IL). After removing unbroken cells and debris by centrifugation at $12,000 \times g$ for 10 min, the membrane fraction was collected by centrifugation at $185,000 \times g$ for one hour and solubilized in S2 buffer (2% Triton X-100 (Anatrace) in PBS (pH 7.4)) for one hour. The soluble fraction was collected after centrifugation at $185,000 \times g$ for one hour and incubated with StrepTactin Sepharose High Performance resin (GE Healthcare, Marlborough, MA) for 30 min using a batch procedure. The resin was transferred into a gravity column (Bio-rad, Hercules, CA) and washed with 10 column volumes of W buffer (100 mM Tris-HCl; pH 8.0, 150 mM NaCl, 1 mM EDTA, and 0.5 mM DDM), and pdP2X7cryst-EGFP was eluted with E buffer (100 mM Tris-HCl; pH 8.0, 150 mM NaCl, 1 mM EDTA, 2.5 mM desthiobiotin, 15% glycerol, and 0.5 mM DDM). The N-terminal EGFP and the strep tag was removed by incubating the P2X7cryst-EGFP with human thrombin (HTI; 1/30 w/w) overnight. P2X7$_{cryst}$ was isolated by size exclusion chromatography using Superdex 200 (GE Healthcare) in C buffer (50 mM Tris-HCl; pH 7.4, 150 mM NaCl, 15% Glycerol, and 0.5 mM DDM). The peak fractions were pooled, concentrated to 10 mg/mL, and used for crystallization after ultracentrifugation at $265,000 \times g$ for 20 min. All purification steps were carried out at 4°C or on ice.

## Crystallization

P2X7$_{cryst}$ was crystalized using the hanging drop vapor diffusion method by mixing 1:1 (v/v) ratio of protein and reservoir solutions at 4°C. The apo crystals appeared in two weeks and were fully grown in 3–4 months using a reservoir solution containing 100 mM HEPES (pH 7.0), 100 mM NaCl, 4%

ethylene glycol, 15% glycerol, 29% PEG-400, 0.1 mg/mL lipid mixture (60% POPE, 20% POPG, and 20% cholesterol). The P2X7$_{cryst}$ crystals with antagonists reached their maximum sizes in about two months using reservoir solutions containing 100 mM HEPES or Tris (pH 6.0–7.5), 100 mM NaCl, 4% ethylene glycol, 15% glycerol, 27–32% PEG-400 or 31–36% PEG-300, 0.1 mg/mL lipid mixture (60% POPE, 20% POPG, and 20% cholesterol), and 1 mM P2X7 antagonists (AZ10606120, JNJ-47965567, A804598, GW791343 (Tocris Biosciences, UK), and A740003 (Sigma Aldrich, St. Louis, MO)). For heavy atom derivatization, A740003-bound P2X7$_{cryst}$ crystals were soaked into a reservoir solution containing 1 mM K2IrCl6 and 0.5 mM DDM overnight at 4°C. For determination of the ATP/ A804598-bound P2X7 structure, crystals of P2X7$_{cryst}$ were grown in 33% PEG-400, 100 mM MES (pH 6.5), 100 mM NaCl, 5% Glycerol, and 1 mM A804598 for 1 month at 4°C. Cubic-shaped crystals were soaked with 1 mM ATP overnight. Crystals were flash-frozen in liquid nitrogen for data collection.

## Structure determination

X-ray diffraction data were collected using synchrotron radiation at the Cornell High Energy Synchrotron Source (beamline F1) and Advanced Photon Source at Argonne National Laboratory (beamlines 24ID-C and E). The following X-ray wave length was used for each data set: Apo: 1.1051 Å (24ID-C); A740003: 0.9792 Å (24ID-E); A804598: 0.9774 Å (F1); AZ10606120: 0.9782 Å (F1); GW791343: 0.9782 Å (F1); JNJ47965567: 0.9774 Å (F1); A740003/K$_2$IrCl$_6$: 1.1051 Å (24ID-C); ATP/A804598: 0.9791 Å (24ID-C). Diffraction data were indexed, integrated, and scaled using XDS (*Kabsch, 2010*), and merged using AIMLESS (*Evans and Murshudov, 2013*) in the CCP4 suite (*Winn et al., 2011*). Low resolution electron density map (~7 Å) of the P4$_2$ crystal was obtained by molecular replacement (RFZ = 7.9, TFZ = 9.3, and LLG = 257) using the zebrafish P2X4 model (PDB code: 4DW0) with the program PHENIX (*Adams et al., 2010*). Initial phase information of the A740003-bound pdP2X7$_{cryst}$ (I2$_1$3 space group) was obtained by MR-SAD with PHENIX using the datasets from Ir-derivatized crystals and the zebrafish P2X4 structure (PDB code: 4DW0) as a searching model. Model building was carried out manually using COOT (*Emsley and Cowtan, 2004*) and the structure was refined using PHENIX. The final model was used as a searching template in molecular replacement for solving the structures of the apo state as well as the antagonist bound forms of pdP2X7$_{cryst}$ with AZ10606120, JNJ47965567, A804598, GW791343. For ATP/A804598-bound form, A804598-bound structure was used as a searching model. Model quality was assessed using PHENIX where stereochemistry and R-values are satisfactory (*Table 2*). Molecular models presented in the figures and videos are created using the PyMOL Molecular Graphics System, Version 1.8 Schrödinger, LLC (*Acquas et al., 1998*). The atomic coordinate files have been deposited in the Protein Data Bank under the accession codes 5U1L (apo), 5U1U (A740003-bound), 5U1V (A804598-bound), 5U1W (AZ10606120-bound), 5U1X (JNJ47965567-bound), 5U1Y (GW791343-bound), and 5U2H (ATP/ A804598-bound).

## YO-PRO-1 uptake assay

Human embryonic kidney (HEK293) cells were maintained in DMEM medium (Thermo Fisher Scientific) supplemented with 10% FBS (Atlanta Biologicals, Flowery Branch, GA), and 10 µg/mL gentamicin (Thermo Fisher Scientific). HEK293 cells were split into six well plates at a cell density of $2.0 \times 10^6$ cells/well and incubated overnight. Three hours after transfection with 2 µg of the full length pdP2X7 in pIM2 vector (IRES-mCherry; modified from pIE2 vector) using jetPRIME reagent (Polyplus-transfection, France), cells were trypsinized, transferred into poly-D-Lysine coated black-walled 96 well plates (Corning, Corning, NY) at $7.5 \times 10^4$ cells/well, and incubated for 24 hr. Cells were washed with assay buffer (2 mM KCl, 0.1 mM CaCl$_2$, 13 mM Glucose, 147 mM NaCl, 10 mM HEPES (pH 7.3)) and were incubated with 5 µM YO-PRO-1 Iodide (Thermo Fisher Scientific), in the presence or absence of antagonists at multiple concentrations at 37°C for 10 min. Upon application of 1 mM ATP (final), uptake of YO-PRO-1 was recorded with 1 min intervals by following the fluorescence change using a Synergy HT multi-detection microplate reader (Bio-Tek, Winooski, VT; Ex: 485 nm/20, Em:528 nm/20, sensitivity 60). To obtain the dose dependent uptake inhibition by antagonists, the initial rates of YO-PRO-1 uptake were plotted against multiple concentrations of the P2X7 specific drugs. The inhibition curves from five independent measurements were fitted to the Hill equation using Origin software (Originlab, Northampton, MA) to determine the IC$_{50}$ values.

**Table 2.** Data collection and refinement statistics.

| | pdP2X7$_{cryst}$ | pdP2X7$_{cryst}$-JNJ47965567 | pdP2X7$_{cryst}$-A740003 | pdP2X7$_{cryst}$-A804598 | pdP2X7$_{cryst}$-AZ10606120 | pdP2X7$_{cryst}$-GW791343 | pdP2X7$_{cryst}$-ATP/A804598 |
|---|---|---|---|---|---|---|---|
| **Data collection** | | | | | | | |
| Space group | $I2_13$ | $I2_13$ | $I2_13$ | $I2_13$ | $I2_13$ | $I2_13$ | $P2_13$ |
| Cell dimensions | | | | | | | |
| $a, b, c$ (Å) | 169.1, 169.1, 169.1 | 169.3, 169.3, 169.3 | 169.6, 169.6, 169.6 | 170.4, 170.4, 170.4 | 170.7, 170.7, 170.7 | 169.7, 169.7, 169.7 | 167.6, 167.6, 167.6 |
| $\alpha, \beta, \gamma$ (°) | 90, 90, 90 | 90, 90, 90 | 90, 90, 90 | 90, 90, 90 | 90, 90, 90 | 90, 90, 90 | 90, 90, 90 |
| Resolution (Å) | 45.2–3.40 (3.52–3.40)* | 45.3–3.20 (3.31–3.20)* | 42.4–3.61 (3.73–3.61)* | 42.6–3.40 (3.52–3.40)* | 42.7–3.50 (3.63–3.50)* | 45.4–3.30 (3.42–3.30)* | 46.5–3.90 (4.04–3.90)* |
| $R_{merge}$ | 0.099 (1.80) | 0.12 (0.95) | 0.23 (1.71) | 0.15 (1.49) | 0.17 (2.21) | 0.12 (1.72) | 0.12 (2.17) |
| $I/\sigma$ | 17.9 (1.30) | 13.4 (2.90) | 13.2 (1.65) | 11.0 (1.71) | 11.01 (1.16) | 12.55 (1.34) | 15.7 (1.25) |
| Completeness (%) | 99.8 (98.8) | 100 (100) | 100 (100) | 100 (100) | 100 (100) | 99.8 (99.4) | 99.8 (99.2) |
| Redundancy | 9.8 (9.9) | 10.1 (10.4) | 11.0 (11.0) | 10.1 (10.0) | 10.1 (9.9) | 10.1 (10.2) | 10.0 (10.3) |
| **Refinement** | | | | | | | |
| Resolution (Å) | 45.2–3.40 | 45.3–3.20 | 42.4–3.61 | 42.6–3.40 | 42.7–3.50 | 45.4–3.30 | 46.5–3.90 |
| No. reflections | 11,197 | 13,482 | 9,510 | 11,474 | 10,598 | 12,383 | 14,510 |
| $R_{work}/R_{free}$ (%) | 24.2/26.3 | 22.3/26.7 | 23.2/26.0 | 25.0/27.3 | 24.5/26.8 | 24.0/27.1 | 33.4/38.6 |
| No. atoms | 2304 | 2473 | 2384 | 2439 | 2454 | 2382 | 4116 |
| Protein | 2276 | 2382 | 2307 | 2359 | 2367 | 2313 | 3967 |
| Ligand/ion | 28 | 91 | 77 | 80 | 87 | 69 | 149 |
| B-factors | 124.4 | 103.6 | 97.8 | 97.5 | 114.9 | 119.0 | 149.3 |
| Protein | 123.8 | 102.9 | 97.1 | 96.8 | 114.5 | 118.5 | 149.1 |
| Ligand/ion | 171.1 | 124.1 | 120.6 | 118.3 | 128.0 | 137.4 | 153.2 |
| R.M.S deviations | | | | | | | |
| Bond lengths (Å) | 0.002 | 0.004 | 0.003 | 0.003 | 0.004 | 0.004 | 0.001 |
| Bond angles (°) | 0.49 | 0.69 | 0.62 | 0.64 | 0.68 | 068 | 0.45 |

*Highest resolution shell is shown in parenthesis.

## Cell line generation

HEK293 (CRL-1573) cell lines were purchased from the American Type Culture Collection (ATCC, Manassas, VA), and therefore were not further authenticated. The mycoplasma contamination test was confirmed to be negative at ATCC.

## Data analysis

Dose response curves of the YO-PRO-1 uptake experiments were fitted with either competitive or non-competitive inhibition models using Origin 6.0 software (OriginLab) as previously described (*Kenakin, 2006*). For the competitive antagonism model, we used the equation:

$$Response = \frac{[A]/KA}{[A]KA(1+\tau)+[B]/KB+1}$$

where $[A]$, $[B]$ are the concentrations of BzATP and antagonists, respectively; $K_A$ and $K_B$ are the equilibrium dissociation-constant of BzATP and antagonists, respectively. Dose response curves without antagonist were fitted with this equation, which gives the values $K_A$ = 28 μM, and $\tau$ = 0.031. $K_B$ was then determined using the dose response curves in the presence of antagonists. The resulting $K_B$

value for each antagonist was; JNJ: 1.7 nM; A80: 15 nM; A74: 24 nM; AZ10: 56 nM; GW: 3.0 μM. For the non-competitive inhibition model, we used the equation:

$$Response = ([A]^n \tau^n Emax)/([A]^n \tau^n + ([A](1+\alpha[B]/K_B) + K_A[B]/K_B + K_A)^n$$

where n is the Hill coefficient, and Emax is the maximum initial rate. $K_A$, $\tau$, n values were determined using the data without antagonists ($K_A$ = 220 μM, $\tau$ = 10, and n = 2.6) and the rest of the parameters were determined by curve fitting of data with different concentrations of antagonists ($\alpha$ = 1 and $K_B$ value for each antagonist was: JNJ: 0.14 μM; A80: 0.57 μM; A74: 1.2 μM; AZ10: 3.4 μM; GW: 266 μM). Schild regression analysis was performed as described previously (*Wyllie and Chen, 2007*). Briefly, concentrations of BzATP at half maximal initial rates of YO-PRO-1 uptake in the presence ($EC_{50}'$) or absence ($EC_{50}$) of antagonists were determined by fitting the dose responses with the Hill equation. Dose ratio $r=EC_{50}'/EC_{50}$ was calculated for various antagonist concentrations and $r-1$ values were plotted against the antagonist concentrations in log scale to obtain Schild plots.

## Ligand-binding experiment

GFP fused pdP2X7$_{cryst}$ (P2X7 GFP) was purified in a buffer containing 150 mM NaCl, 50 mM Tris-HCl (pH 7.4), 15% glycerol, and 0.5 mM DDM as described in "Expression and purification." GFP-tagged pdP2X7$_{cryst}$, which is substantially more stable than pdP2X7$_{cryst}$, was used in this experiment as it does not interfere with the fluorescence properties of Alexa-ATP (*Figure 3—figure supplement 5B*). P2X7-GFP (100 μM) was pre-incubated with each P2X7 specific antagonist (100 μM) for 30 min at room temperature. P2X7 GFP was then incubated with 10 μM ATP-Alexa 647 (Thermo Fisher Scientific) at 30°C for 10 min, which was required to obtain a stable background prior to the fluorescence measurement. Fluorescence anisotropy was measured at 30°C using FluoroMax four fluorimeter (Horiba,Edison, NJ) with excitation and emission wavelengths of 590 nm and 670 nm, respectively. For binding competition experiments, various concentrations of ATP ranging from 10 μM to 10 mM (pH was adjusted to 7.0 with NaOH) were added from 100X solutions. Fluorescence anisotropy $\langle r \rangle$ was defined as:

$$\langle r \rangle = \frac{IVV - G \ast IVH}{IVV + 2 \ast G \ast IVH}$$

where $I_{VV}$ and $I_{VH}$ are the fluorescence intensities with the excitation polarizer mounted vertically and the emission polarizer mounted vertically or horizontally, respectively. *G* is defined as:

$$G = \frac{IHV}{IHH}$$

where $I_{HV}$ and $I_{VV}$ are the fluorescence intensities with the excitation polarizer mounted horizontally and the emission polarizer mounted vertically or horizontally, respectively.

## Electrophysiology

HEK293 cells were split onto glass coverslips in six well plates at $1 \times 10^5$ cells/well and incubated at 37°C overnight. Cells were transfected with 1 μg of the full length pdP2X7 (wildtype or mutants) or the full length mP2X4 (wildtype or mutants) in pIE2 vector using FuGENE6 (Promega, Madison, WI). Cells were used 18–32 hr after transfection for measuring the P2X receptor activities using the whole cell patch clamp configuration. Membrane voltage was clamped at −60 mV with an Axopatch 200B amplifier (Molecular Devices, Sunnyvale, CA), currents were filtered at 2 kHz (eight-pole Bessel; model 900BT; Frequency Devices, Ottawa, IL) and sampled at 10 kHz using a Digidata 1440A and pCLAMP 10.5 software (Molecular Devices). The extracellular solution contained 147 mM NaCl, 10 mM HEPES, 13 mM Glucose, 2 mM KCl, 0.1 mM CaCl$_2$, (pH 7.3). The pipette solution contained 147 mM NaCl, 10 mM HEPES, 10 mM EGTA, which was adjusted to pH 7.0 using NaOH. Whole cell configuration was made in an extracellular solution supplemented with 2 mM CaCl$_2$ and 1 mM MgCl$_2$ and the extracellular solutions were rapidly exchanged to the solutions containing desired concentrations of ATP using a computer-controlled perfusion system (RSC-200; Bio-Logic, France). Because pdP2X7 substantially runs up (*Figure 1B and E*), we measured the channel activity after treating the cells with 1 mM ATP for at least 20 s. For testing the effects of P2X7 specific antagonists on pdP2X7, these drugs were incubated with ATP (1 mM) for 1 min. Concentrations of the drugs were: A740003:

600 nM; A804598: 180 nM; AZ10606120: 2.3 µM; GW791343: 50 µM; JNJ47965567: 136 nM. For the cysteine accessibility studies on pdP2X7, 0.1 mM MTS-TPAE (Toronto Research Chemicals, Canada) was perfused for 10 s either in the absence or presence of 1 mM ATP. For probing mP2X4 accessibility in the closed state, 0.1 mM MTS-TPAE was applied for 10 s and application of 10 µM ATP for 1 s was used to measure channel activity. For mP2X4 accessibility in the open state, 5 µM ATP was applied for 9 s and 0.1 mM MTS-TPAE was applied for 3 s. For measuring cysteine accessibility of pdP2X7/Y295C or mP2X4/F296C mutants, cells were treated with 10 mM dithiothreitol (DTT) for 5 min prior to recording. To normalize the channel activities from multiple experiments, the ratio between channel activity before and after MTS-TPAE application was calculated for each cell. Since mP2X4 rapidly runs down, these ratios were further normalized to the control channel activities measured on the same cell in the absence of MTS-TPAE. Stock solutions of 100 mM MTS-TPAE were prepared freshly every 5 hr in water and stored on ice and were diluted to the desired concentrations in the extracellular solution immediately before each recording. All recordings were performed using more than five cells.

## Acknowledgements

We thank the personnel at beamlines 24-ID of the Advanced Photon Source, X-25 of National Synchrotron Light Source, and F1 of the Cornell High Energy Synchrotron Source. We also thank H Lin, G Hollopeter, R Cerione, C Sevier, and K Michalski for discussion.

## Additional information

### Funding

| Funder | Grant reference number | Author |
| --- | --- | --- |
| National Institutes of Health | NS072869 | Toshimitsu Kawate |
| National Institutes of Health | GM114379 | Toshimitsu Kawate |

The funders had no role in study design, data collection and interpretation, or the decision to submit the work for publication.

### Author contributions

AK, Acquisition of data, Analysis and interpretation of data, Drafting or revising the article, Contributed unpublished essential data or reagents; TK, Conception and design, Analysis and interpretation of data, Drafting or revising the article, Contributed unpublished essential data or reagents

### Author ORCIDs

Toshimitsu Kawate, http://orcid.org/0000-0002-5005-2031

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
