## [Decision Letter]

Thank you for submitting your article "Structural basis for subtype-specific inhibition of the P2X7 receptor" for consideration by *eLife*. Your article has been favorably evaluated by Gary Westbrook as the Senior Editor and three reviewers, one of whom, Kenton J Swartz (Reviewer #1), is a member of our Board of Reviewing Editors. The following individuals involved in review of your submission have agreed to reveal their identity: Hiro Furukawa (Reviewer #2) and Mark L Mayer (Reviewer #3).

The reviewers have discussed the reviews with one another and the Reviewing Editor has drafted this decision to help you prepare a revised submission.

Summary:

This is an exciting manuscript describing the first X-ray structure of the P2X7 receptor, a subtype of ATP-activated P2X receptors implicated in a range of interesting biological processes, including pain and apoptosis. Structures were solved in the presence of 5 distinct inhibitors, and for one of them in the combined presence of ATP. These structures show that the drugs bind in a completely unanticipated region of the receptor, in a cleft originating from the external surface of the receptor that resides behind the ATP binding pocket. In the one case when ATP was bound to the receptor along with antagonist, closure of the lateral ATP binding domain occurs similar to what is seen in the original structure of the ATP-bound zfP2X4 receptor, as well as the very recently published ATP-bound P2X3 receptor, but opening the pore is prevented by the presence of the drugs in the cleft, suggesting that it narrows as the channel opens. Interestingly, the drug-binding cleft is essentially absent from P2X3 and 4 receptors, explaining why the drugs studied here are selective for P2X7. The new structures presented are accompanied by extensive functional experiments, including 1) mutagenesis of the drug-binding cleft to confirm the unexpected binding site, 2) Schild analysis demonstrating that the drugs do not work through a competitive mechanism as suggested for 3 of the compounds by earlier work, and 3) accessibility studies with bulky MTS reagents that are consistent with the cleft being wide in the closed state and narrowing upon ATP binding. Overall this is a beautiful study that arrives at interesting and important new conclusions concerning the gating mechanism and mechanism of drug action for the P2X7 receptor. The work could be published anywhere and *eLife* is fortunate to have this one. Revision should be straightforward.

Essential revisions:

The following two concerns could be addressed by careful revision of the manuscript without performing additional experiments, but it is important that they take these issues seriously.

1) Although the location of the bound drug molecules is convincingly demonstrated by Fo-Fc maps, it is far from clear that the current resolution of any structure reported in the paper (3.2 – 3.9 Å) is high enough to support the detailed models in presented in Figure 3 in which side chain interactions with bound ligands are reported. Indeed, the 2Fo-Fc map shown in Figure 2 appears to show tubes of electron density for α-helices, without any side chains. How robust are the chosen side chain rotamers during refinement; is the helical register correct; how do these issues impact the models shown in Figure 3? Also, how dependent on side chain conformations are the HOLE plots and graph presented in Figure 4?

2) The conclusion that the antagonists "allosterically prevent narrowing of the drug-binding pocket and the turret-like architecture during channel opening" is a reasonable hypothesis derived from a combination of functional experiments with MTS reagents, and a comparison with previously reported P2X3 and P2X4 crystal structures, but there is no direct proof of this. In our opinion the authors overstate what their data actually shows. What is lacking is an agonist (ATP) bound P2X7 structure in which the turret has adopted an active compact conformation. They should carefully revise the manuscript to address this, and perhaps explain why this was not done, and an indirect approach of functional assays with Cys mutants and MTS-TPAE adopted.

---

## [Author Response]

*Essential revisions:*

*The following two concerns could be addressed by careful revision of the manuscript without performing additional experiments, but it is important that they take these issues seriously.*

1) Although the location of the bound drug molecules is convincingly demonstrated by Fo-Fc maps, it is far from clear that the current resolution of any structure reported in the paper (3.2 – 3.9 Å) is high enough to support the detailed models in presented in Figure 3 in which side chain interactions with bound ligands are reported. Indeed, the 2Fo-Fc map shown in Figure 2 appears to show tubes of electron density for α-helices, without any side chains. How robust are the chosen side chain rotamers during refinement; is the helical register correct; how do these issues impact the models shown in Figure 3? Also, how dependent on side chain conformations are the HOLE plots and graph presented in Figure 4?

We agree with the reviewers that the resolution of the reported crystal structure is not high enough to determine the accurate distances/angles between the drug-binding residues and the drugs. We therefore avoided describing potential specific-interactions, such as hydrogen bonding. We also performed site-directed mutagenesis for the drug-surrounding residues, where we validated the involvement of these residues in drug-binding. Nevertheless, locations and orientations of the side chains around the drug-binding pocket are well-supported by the reasonable electron density in these regions. To clarify these points, we added 1) a sentence in the main text describing the limitation of our current structures, 2) a new video (Video 1) showing the electron density of the drug-binding residues around one of the antagonists (JNJ47965567), and 3) a new figure (Figure 3—figure supplement 1) that shows the side chain electron density for the residues comprising the turret (β13 and β14) and the α-helix whose helical-register was questioned. We believe that these extra video and figure support the models presented in Figure 3 as well as the HOLE plots and graph presented in Figure 4.

The 2Fo-Fc map shown in Figure 2—figure supplement 2 includes only one short helix (D89-T90-A91-D92-Y93) whose side chain densities are relatively well-defined. Unfortunately, the angle of this figure prevents readers from appreciating how good the side chain densities are, as the point of this figure was to show no obvious density in the drug-binding pocket. The newly added figure (Figure 3—figure supplement 1) should solve this issue. In addition, combining the fact that the helical register in this region is consistent with those of P2X3 and P2X4, miss-assignment of residues in this region is highly unlikely.

We selected the side chain rotamers that fit best in the electron density, which is a standard way of modeling a low resolution structure. While there is a chance that the side chain rotamer is incorrect, a wrongly assigned rotamer should not have major impact on the models shown in Figure 3, as the approximate orientation of each side chain should remain the same. Likewise, the HOLE plots would be only minutely different, if different side chain conformations are used. Indeed, positions of the peptide backbone account for the major differences in the HOLE plots.

*2) The conclusion that the antagonists "allosterically prevent narrowing of the drug-binding pocket and the turret-like architecture during channel opening" is a reasonable hypothesis derived from a combination of functional experiments with MTS reagents, and a comparison with previously reported P2X3 and P2X4 crystal structures, but there is no direct proof of this. In our opinion the authors overstate what their data actually shows. What is lacking is an agonist (ATP) bound P2X7 structure in which the turret has adopted an active compact conformation. They should carefully revise the manuscript to address this, and perhaps explain why this was not done, and an indirect approach of functional assays with Cys mutants and MTS-TPAE adopted.*

We agree with the reviewers and rephrased the stronger statement about the conformational changes. We also added a description about why we are short in obtaining an ATP-bound P2X7 structure in the Discussion of the revised manuscript.